# Substantial acetylcholine reduction in multiple brain regions of *Mecp*2-deficient female rats and associated behavioral abnormalities

Hiroyasu Murasawa[1,2], Hiroyuki Kobayashi[1,2], Jun Imai[2], Takahiko Nagase[2], Hitomi Soumiya[1], Hidefumi Fukumitsu[1]*

1 Laboratory of Molecular Biology, Department of Biofunctional Analysis, Gifu Pharmaceutical University, Gifu, Japan, 2 Hashima Laboratory, Nihon Bioresearch Inc, Gifu, Japan

* hfukumi@gifu-pu.ac.jp

## Abstract

Rett syndrome (RTT) is a neurodevelopmental disorder with X-linked dominant inheritance caused mainly by mutations in the methyl-CpG-binding protein 2 (*MECP2*) gene. The effects of various *Mecp2* mutations have been extensively assessed in mouse models, but none adequately mimic the symptoms and pathological changes of RTT. In this study, we assessed the effects of *Mecp2* gene deletion on female rats (*Mecp2*[+/−]) and found severe impairments in social behavior [at 8 weeks (w), 12 w, and 23 w of age], motor function [at 16 w and 26 w], and spatial cognition [at 29 w] as well as lower plasma insulin-like growth factor (but not brain-derived neurotrophic factor) and markedly reduced acetylcholine (30%–50%) in multiple brain regions compared to female *Mecp2*[+/+] rats [at 29 w]. Alternatively, changes in brain monoamine levels were relatively small, in contrast to reports on mouse *Mecp2* mutants. Female *Mecp2*-deficient rats express phenotypes resembling RTT and so may provide a robust model for future research on RTT pathobiology and treatment.

## Introduction

Rett syndrome (RTT) (OMIM 312750) is an X-linked progressive neurodevelopmental disorder with an incidence of about 1:10,000 among newborn females, making it the second most frequent cause of female mental retardation after Down syndrome [1]. It also shortens life expectancy by ~30–40 years in females, while males usually demonstrate early onset and die shortly after birth. Mutations in the gene encoding transcription regulator methyl-CpG-binding protein 2 (*MeCP2*) are identified in more than 95% RTT patients [2, 3]. Females with RTT appear to develop normally until 6 to 18 months after birth, but then begin to exhibit deficits in cognitive function, motor function, and sociality, with continued developmental regression characterized by a loss of acquired communication and purposeful hand skills [4, 5].

Over 300 *MECP2* mutations and genomic abnormalities have been documented in RTT patients as well as in other mental health disorders, including intellectual disability, autism spectrum disorder, bipolar disorder, and schizophrenia [6]. Based on these genetic studies, several mouse models with various *Mecp2* mutations have been developed and studied over the past

**Funding:** The funder, Nihon Bioresearch Inc, provided support in the form of salaries for authors: HM, HK, JI, and TN, but did not have any additional role in the study design, data collection and analysis, decision to publish, or preparation of the manuscript. No additional external funding was received for this study.

**Competing interests:** Authors HM, HK, JI, and TN are paid employees of Nihon Bioresearch Inc. There are no patents, products in development or marketed products associated with this research to declare. This does not alter our adherence to PLOS ONE policies on sharing data and materials.

**Abbreviations:** BDNF, Brain-derived neurotrophic factor; ChAT, Choline acetyltransferase; CSF, Cerebrospinal fluid; DA, Dopamine; DBH, Dopamine β-hydroxylase; DOPAC, 3,4-dihydroxyphenylacetic acid; EDTA, Ethylenediaminetetraacetic acid; ELISA, Enzyme-linked immunosorbent assay; 5-HT, Serotonin; 5-HIAA, 5-hydroxy indole-3-acetic acid; GABA, γ-aminobutyric acid; GFAP, Glial fibrillary acidic protein; IGF-1, Insulin-like growth factor 1; LC, Locus coeruleus; MECP2, Methyl-CpG-binding protein 2; MHPG, 3-methoxyphenyl-4-hydroxy-glycol; MWM, Morris water maze; NE, Norepinephrine; NMDA, N-methyl-D-aspartate; HPLC, High-performance liquid chromatography; HPLC-ECD, High-performance liquid chromatography with an electrochemical detector; HVA, Homovanillic acid; RTT, Rett syndrome; SD, Sprague Dawley; SE, Standard error of the mean; TH, Tyrosine hydroxylase; TPH, Tryptophan hydroxylase; w, Weeks.

two decades [7, 8]. One of the most striking findings is that many phenotypes reported in *Mecp2* gene null mutants [9] are also observed in mouse mutants generated using a post-mitotic neuron-specific knockout strategy [10], consistent with postnatal neurodevelopmental regression as a predominant symptom. Moreover, some of these phenotypes can be normalized by re-expression of the *Mecp2* gene after birth [11–13]. This potential postnatal reversibility of RTT phenotypes has prompted research on therapeutics to compensate for *Mecp2* and associated deficiencies [14], but there has been no successful clinical translation of these findings.

While mouse models allow for convenient genetic manipulation and large-scale breeding, small body and brain sizes limit critical research methods such as region-specific *in vivo* electrophysiology and neurochemistry (e.g., microdialysis). Further, some complex and sophisticated behaviors are not observed in mice [15, 16], which may make it difficult to capture RTT-related regressive pathologies. Considering these shortcomings, a rat model was recently developed to investigate the effects of *Mecp2* deficiency on brain development, cognition, and behavior [17–19].

Since human RTT is limited to females, it is preferable to conduct all animal model experiments on females, although few studies have examined female mutant mice exclusively, likely because of the more delayed and complex phenotypic progression associated with cellular mosaicism [8]. To further explore the pathological progression associated with *Mecp2* gene deficiency, we examined differences in spatial learning and memory, histopathology, and regional neurotransmitter levels between wild-type (*Mecp2*$^{+/+}$) and *Mecp2*$^{+/-}$ female rats. These studies revealed progressive deficits in social behavior and motor function as well as impaired spatial learning and memory. Further, regional acetylcholine levels were substantially reduced, suggesting that disrupted cholinergic transmission or metabolism may contribute to the behavioral and cognitive abnormalities of RTT.

## Materials and methods

All experimental protocols were approved by the Institutional Animal Care and Use Committee of Nihon Bioresearch Inc. (protocol number: 201601), and performed in compliance with the Guidelines for Management and Welfare of Experimental Animals of both Nihon Bioresearch and the National Institutes of Health. All efforts were made to minimize animal suffering and to reduce the number of animals used. We adopted the following protocol: if a rat showed any symptoms of suffering, such as abnormal breathing and pulse, decreased body temperature, lying down with less response to external stimuli, or showed rapid weight loss (>20% in several days), we would take euthanasia measures by exsanguination with the opening of the abdominal aorta under isoflurane anesthesia. None of the rats, however, showed any such symptoms. We therefore used all the datasets without any exclusion throughout this study.

### Animals

Female *Mecp2* gene mosaic heterozygotic Sprague Dawley (SD) rats (*Mecp2*$^{+/-}$) were obtained from SAGE Labs (part of Horizon Discovery, UK). The genotype of these animals is guaranteed by SAGE Labs. The zinc-finger nuclease technology was used to establish an animal model for generating a 71-base pair deletion in exon 4 [18]. Female wild-type control SD rats (*Mecp2*$^{+/+}$) were obtained from Charles River Laboratories (Yokohama, Japan) at 6 weeks (w) of age and housed under the same conditions as mutants (n = 6 for each genotype, respectively). Briefly, all animals were housed in an animal room with temperature maintained at 18.0°C to 28.0°C, relative humidity at 30.0%–80.0%, and a 12-hour/12-hour light/dark cycle (lighting: 6:00 a.m. to 6:00 p.m.) with filtered fresh air changes 12 times per hour. The animals

were housed individually in plastic cages (W: 310 × D: 360 × H: 175 mm) lined with autoclaved paper bedding and allowed free access to food and water.

## Study schedule

Behavioral test batteries are described in the order conducted. The social interaction test was performed at 8, 12, and 23 w of age, the locomotor activity test at 16 and 23W, rotarod performance test at 26W, and Morris water maze (MWM) test at 29 w (Fig 1). One week after the final MWM tests, blood was collected from the postcava into 1-mL disposable polypropylene syringe (Terumo Corporation, Tokyo, Japan) using a 23G needle (Terumo Corporation) under isoflurane anesthesia (Isoflurane Inhalation Solution [Pfizer], Mylan Inc., Osaka, Japan). The animals were then euthanized by bleeding, and the brain was collected. Half of the brain was fixed in 4% paraformaldehyde phosphate buffer solution and processed for immunohistology while the other half was used for measurement of brain neurotransmitters.

## Behavioral test batteries

**Social interaction test.** The social interaction test was performed in a clean plastic cage of the same type used for animal housing. Two female $Mecp2^{+/-}$ or $Mecp2^{+/+}$ rats that had never

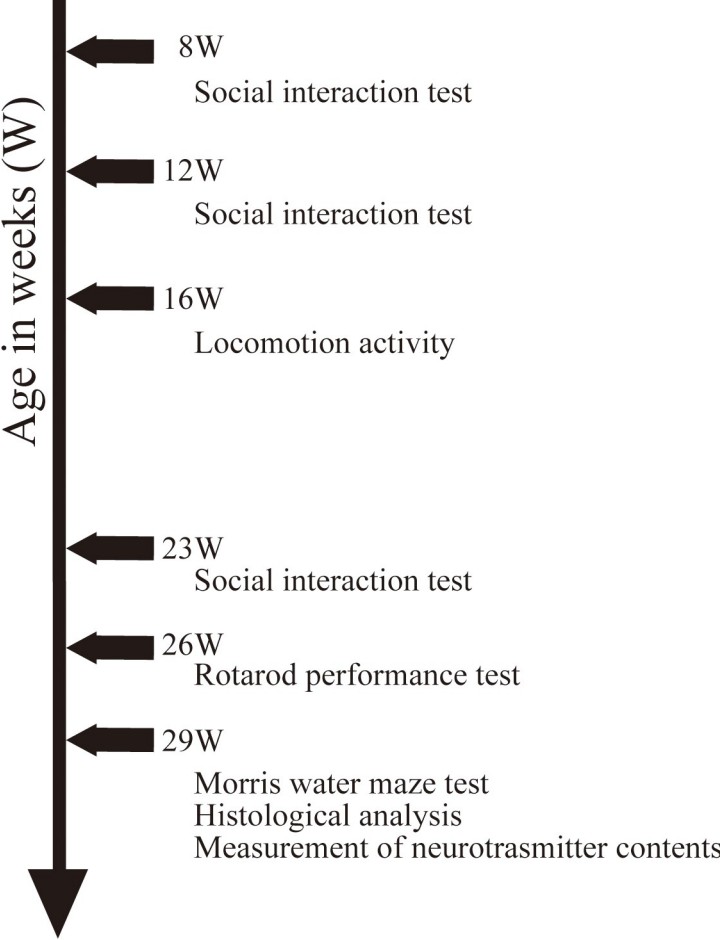

**Fig 1. Experimental schedules.** This figure describes the experimental schedules for behavioral test batteries and tissue sample preparations.

been in contact were placed together in the cage, and social behaviors were observed for 20 minutes under interior light through a transparent cage top with air vents. Frequencies and accumulated durations of contact behaviors (genital investigation, sniffing, and social grooming) and independent self-grooming were measured separately with a counter and a stopwatch.

**Locomotor activity test.** Locomotor activity over 2 consecutive 24-periods with 12-hour light and dark cycles (48 h) was analyzed at 30-minute intervals in the home cage using a computerized system (Multi digital 32 port count system, Neuroscience, Inc. Osaka, Japan).

**Rotarod test.** To evaluate motor coordination, the animal was placed on a rotarod (Rotarod, ENV-577, Med. Associates, Inc., St. Albans, VT, U.S.A). The rotation speed was gradually accelerated from 4 to 40 rpm and the time (in seconds) and speed at which the rat fell off were measured within 5 minutes.

**Morris water maze (MWM) test.** Spatial learning and memory were assessed in the MWM. The MWM apparatus consisted of a gray vinyl chloride circular pool (diameter: 148 cm, height: 44 cm) filled with water (17˚C to 18˚C) up to a height of about 32 cm so that a clear acrylic escape platform (12-cm in diameter) was submerged approximately 2 cm underneath the surface. The pool was divided onto four equal quadrants, with the escape platform located in the center of the fourth quadrant (see Fig 6A). In the acquisition phase, the animal was randomly placed in the water at a wall position between 'a' and 'e' (see Fig 6A) with the head facing the wall, and swimming behavior was recorded by a video camera set above the pool and displayed on a TV monitor. In acquisition trials, time to reach the platform (escape latency in seconds) and swimming distance (cm) were analyzed using a video tracking system (Etho Vision XT, Noldus Information Technology Inc., PA Wageningen, Netherlands) as indices of spatial learning. Acquisition trials were performed twice a day for 4 days (8 times in total). If the animal did not find the platform within 90 s, latency was recorded as 91 s and it was gently guided onto the platform and allowed to remain for 30 s. One day following the final acquisition trial, animals were subjected to a probe trial for spatial memory in which the escape platform was removed. The animal was released from c and allowed to swim freely for 90 s. Time in the former platform quadrant (quadrant 4) and number of platform position crossings were recorded as indices of spatial memory.

## High-performance liquid chromatography (HPLC) analysis of biogenic amine levels

Regional neurotransmitter levels were measured in excised brain tissue by isocratic HPLC separation and electrochemical detection. Briefly, the animals were euthanized by bleeding from the postcava under isoflurane anesthesia one week after the MWM test, and the brain tissues were collected in ice-cooled physiological saline. Using a metallic zinc brain matrix (ASI Instruments Inc. Warren, MI, U.S.A), each hemisphere was sliced into 2-mm thick coronal sections and divided into frontal cortex, amygdala, hippocampus, caudate nucleus, thalamus, hypothalamus, and medulla oblongata according to the rat brain atlas of Paxinos and Watson [20]. Each region was collected in a separate container, weighed, frozen in liquid nitrogen, and stored at −90˚C to −70˚C in a deep freezer (ULT-1186-3SI-A36, Thermo Electron Corporation Co., Ltd., Tokyo, Japan) until neurotransmitter measurements.

The frozen brain tissue was homogenized in 0.2 mol/L perchloric acid and centrifuged at 20227g and 4˚C for 5 minutes. The following neurotransmitters and metabolites were measured from the supernatant using the HPLC-ECD system (Eicom Corporation, Kyoto, Japan): dopamine (DA) and its metabolites 3,4-dihydroxyphenylacetic acid (DOPAC) and homovanillic acid (HVA), serotonin (5-HT) and its metabolite 5-hydroxy indole-3-acetic acid (5-HIAA),

norepinephrine (NE) and its metabolite 3-methoxyphenyl-4-hydroxy-glycol (MHPG), gluta-mate, γ-aminobutyric acid (GABA), and acetylcholine as follows.

The levels of monoamines (DA, 5-HT, and NE) and monoamine metabolites (DOPAC, HVA, 5-HIAA, and MHPG) in the brain tissue were estimated using an HPLC system equipped with a C18 reverse-phase column (Eicompak SC-5ODS, 150 mm × 3.0 ID, Eicom Corporation, Kyoto, Japan), guard column (PC-04, Eicom Corporation, Kyoto, Japan), and column thermostat (ATC-700) set to 25˚C. Molecules were separated by a mixture of 0.1 mol/ L acetic acid–citrate buffer (pH 3.5), methanol, 100 mg/mL sodium 1-octanesulfonate solution, and 5 mg/mL ethylenediaminetetraacetic acid (EDTA)-2Na solution (820:180:2.2:1) under the control of a pump system (EP-700) at a flow rate of 0.5 mL/min. For each measurement, 20 μL of the brain lysate supernatant was injected automatically using an automated sample injector (M-500). An electrochemical detector (ECD-700) with a graphite electrode (WE-3G), gasket, and Ag/AgCl reference electrode (RE-500) set to +750 mV was used for quantification.

Glutamate level was measured using an HPLC system equipped with a C18 reverse-phase column (Eicompak E-GEL, 150 mm × 4.6 ID, Eicom Corporation, Kyoto, Japan), enzyme col-umn (E-EMZYMPAK, 4 mm × 3.0 ID, Eicom Corporation, Kyoto, Japan), guard column (PC-03, Eicom Corporation, Kyoto, Japan), and column thermostat (ATC-700) set to 33˚C. A mix-ture of 60 mM ammonium chloride–ammonia solution (pH 7.2), hexadecyltrimethylammo-nium bromide, and 5 mg/mL EDTA-2Na solution (1000 mL:250 mg:0.01 mL) was used as the mobile phase under the control of the pump system (EP-700) at 0.37 mL/min. For each mea-surement, 20 μL of the brain lysate supernatant was injected automatically and the level mea-sured using the electrochemical detector (ECD-700) with a platinum electrode (WE-PT), gasket, and Ag/AgCl reference electrode (RE-500) set at +450 mV.

GABA level was measured using an HPLC system equipped with a C18 reverse-phase col-umn (Eicompak FA-3ODS, 50 mm × 3.0 ID, Eicom Corporation, Kyoto, Japan), guard column (PC-03, Eicom Corporation, Kyoto, Japan), and column thermostat (ATC-700) set to 40˚C. A mixture of 0.1 mol/L phosphate buffer solution (pH 6.0), acetonitrile, and 5 mg/mL EDTA-2Na solution (500 mL:500 mL:1 mL) was used as the mobile phase under the control of the pump system (EP-700) at a flow rate of 0.5 mL/min. For each measurement, 20 μL sample was injected automatically using the automated sample injector (M-500) and the level measured using the electrochemical detector (ECD-700) with a graphite electrode (WE-GC), gasket, and Ag/AgCl reference electrode (RE-500) set at +600 mV.

Acetylcholine level was measured using an HPLC system equipped with a C18 reverse-phase column (Eicompak AC-GEL, 150 mm × 4.6 ID, Eicom Corporation, Kyoto, Japan), enzyme column (E-EMZYMPAK, 4 mm × 1.0 ID, Eicom Corporation, Kyoto, Japan), guard column (PC-03, Eicom Corporation, Kyoto, Japan), and column thermostat (ATC-700) set to 33˚C. A mixture of 0.1 mol/L potassium bicarbonate solution, sodium decanesulfonate, and 5 mg/mL EDTA-2Na solution (1000 mL:400 mg:0.01 mL) was used as the mobile phase under the control of the pump system (EP-700) at a flow rate of 0.15 mL/min. For each measurement, 20 μL sample was injected automatically using the automated sample injector (M-500) and the level measured by the electrochemical detector (ECD-700) with a platinum electrode (WE-PT), gasket, and Ag/AgCl reference electrode (RE-500) set at +450 mV.

## Tissue preparation and immunohistochemical analysis

The fixed brain tissue was embedded in paraffin using the routine method of our testing facil-ity and cut into 5-μm thick sections. Every third section was processed for immunohistochem-ical staining with anti-glial fibrillary acidic protein (GFAP) antibody (×2000, Abcam, ab 7260). The three to five sections of choice were those closest to dorsal–ventral level interaural 3.20

mm (for frontal cortex) and -3.80 mm (for the hippocampus, hypothalamus, and amygdala) according to Paxinos and Watson [20]. Immunostained sections of the frontal cortex and hippocampus were photographed using an optical microscope with a digital camera, and GFAP-positive cells were counted using WinROOF V7.0 software.

## Hematological examination

Plasma was collected by centrifugation of the collected blood (at 4°C and 2150×g for 15 minutes). Plasma insulin-like growth factor 1 (IGF-1) was estimated using the Quantikine® ELISA kit (Mouse/Rat IGF-I immunoassay, Cosmo Bio Co., Ltd., Tokyo, Japan) and plasma BDNF using the RayBio® Rat BDNF ELISA Kit (Ray Biotech. Inc., Peachtree Corners, GA, U.S.A).

## Statistical analysis

Data are presented as the mean ± standard error of the mean (SE). All statistical analyses were performed using GraphPad Prism version 9.1.2 for Windows (GraphPad Software Inc., San Diego, CA, U.S.A, www.graphpad.com), according to Prism9 Statistics Guide (https://www.graphpad.com/guides/prism/latest/statistics/index.htm). All details of statistics have been indicated in figure legends. Because the sample size for each experiment was small, it was not appropriate to consider that the sample was normally distributed. Therefore, when comparing two independent groups, the unpaired two-tailed Mann–Whitney U-test was used. For the analysis of statistical significance in comparisons involving more than two groups, Kruskal–Wallis with Dunn's *post hoc* multiple comparisons test was used. When comparing three or more matched groups, the Friedman test was used. In all cases, statistical significance was assessed with a 95% confidence interval; therefore, $p < 0.05$ was considered significant.

# Results

## Impaired social behaviors in *Mecp2*-deficient female rats

We first analyzed the effect of *Mecp2* gene deficiency on rat social behaviors at 8, 12, and 23 w. The frequency and duration of contact behaviors gradually increased with age among female *Mecp2*$^{+/+}$ rats but gradually declined with age among female *Mecp2*$^{+/−}$ rats, and both parameters were significantly lower compared to female *Mecp2*$^{+/+}$ rats by 23 w (Fig 2A and 2B). We next analyzed self-grooming behavior during the social interaction test as higher frequency and longer duration are considered endophenotypic hyper-repetitive behaviors of autism spectrum disorder [21–23] and male *Mecp2* mutant mice [24–26]. The frequencies (Fig 3A) of self-grooming were significantly reduced and the duration (Fig 3B) tended to be reduced but not significant in female *Mecp2*$^{+/−}$ rats. Thus, the deficiency in *Mecp2* impaired social behaviors with reduced self-grooming.

## Impaired spontaneous locomotor activity and motor coordination in *Mecp2*-deficient female rats

To evaluate the effects of *Mecp2* deficiency on motor function, we first analyzed spontaneous locomotor activity over 2 consecutive 24-periods with 12-h/12-h light/dark cycles at 16 w and 23 w. Spontaneous locomotor activity was significantly or tended to be lower among female *Mecp2*$^{+/−}$ rats compared to female *Mecp2*$^{+/+}$ rats (Fig 4) during both light and dark phases. In addition, judging from the locomotor activity counts every 30 minutes, *Mecp2* deficiency did not seem to affect circadian rhythms because female *Mecp2*$^{+/-}$ rat shows a tendency to decrease activity over the entire measurement time, not at any specific time point (S1 Fig). To assess motor coordination, animals were tested on the accelerating rotarod at 26 w. Both rotation

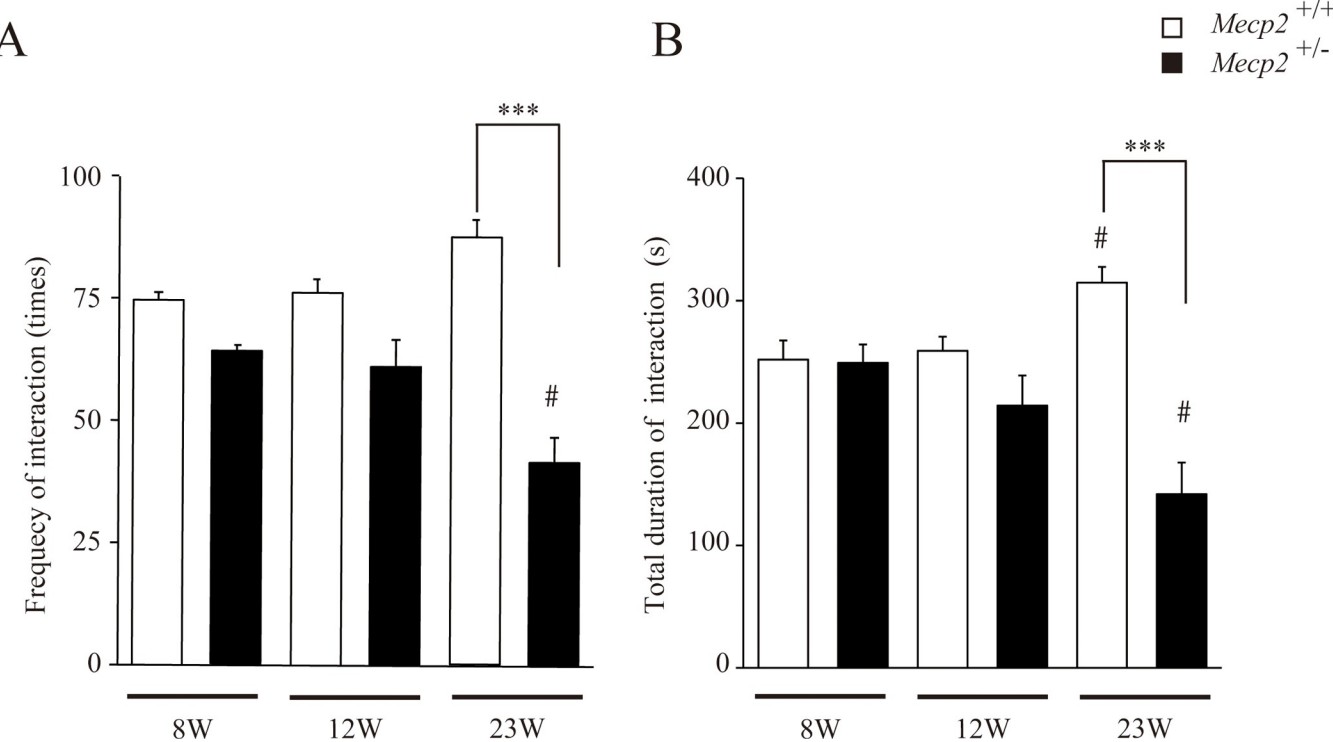

**Fig 2. Age-dependent alterations in social behaviors among female *Mecp2*<sup>+/−</sup> rats.** Female *Mecp2*$^{+/−}$ rats exhibited progressively less frequent social interactions with age, while social interactions increased with age among female *Mecp2*$^{+/+}$ (wild-type) rats. Bar graphs indicate frequency of contact behaviors (A) and total duration of interactions (B) during a 20-min social interaction test (mean ± SE; n = 6 rats per genotype). $^{#}p < 0.05$, and $^{***}p < 0.001$ vs. same genotyped female *Mecp2*$^{+/−}$ rats at 8 weeks (w) (Friedman test), and the age-matched female *Mecp2*$^{+/+}$ rats at 23 w of age (Kruskal–Wallis with Dunn's *post hoc* multiple comparisons test), respectively.

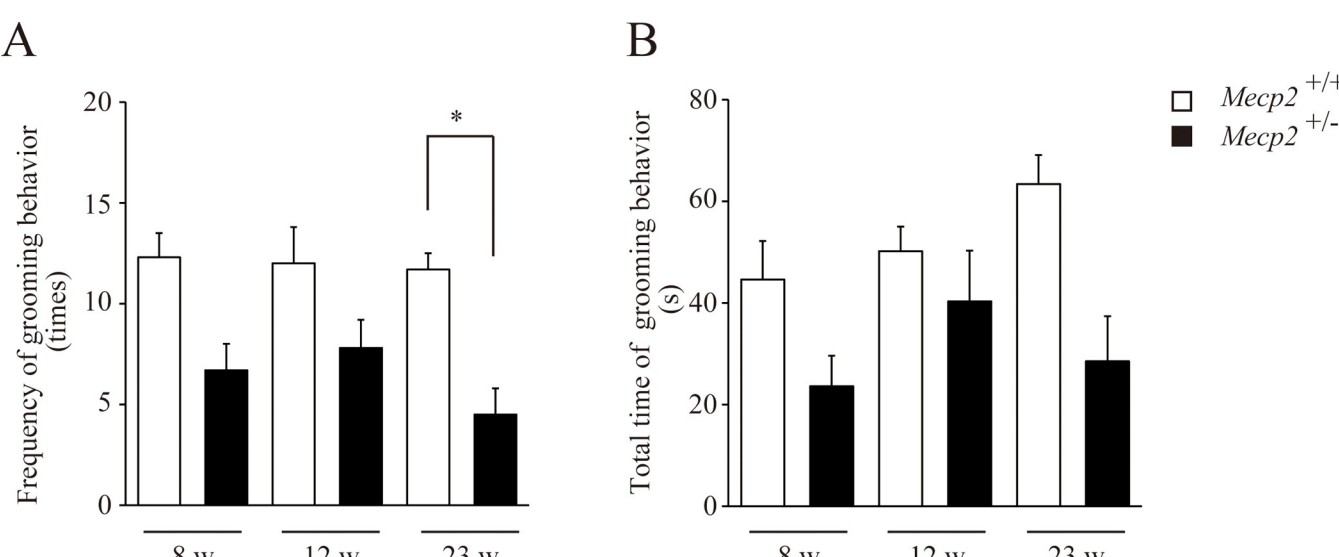

**Fig 3. Age-dependent changes in self-grooming behavior among female *Mecp2*<sup>+/−</sup> rats.** Both frequency (A) and total time (B) of self-grooming tended to be lower in female *Mecp2*$^{+/−}$ rats compared to female *Mecp2*$^{+/+}$ rats during the 20-min social interaction test at 8, 12, and 23 w of age (mean ± SE; n = 6 rats per genotype). $^{*}p < 0.05$ vs. age-matched female *Mecp2*$^{+/+}$ rats (Kruskal–Wallis with Dunn's *post hoc* multiple comparisons test).

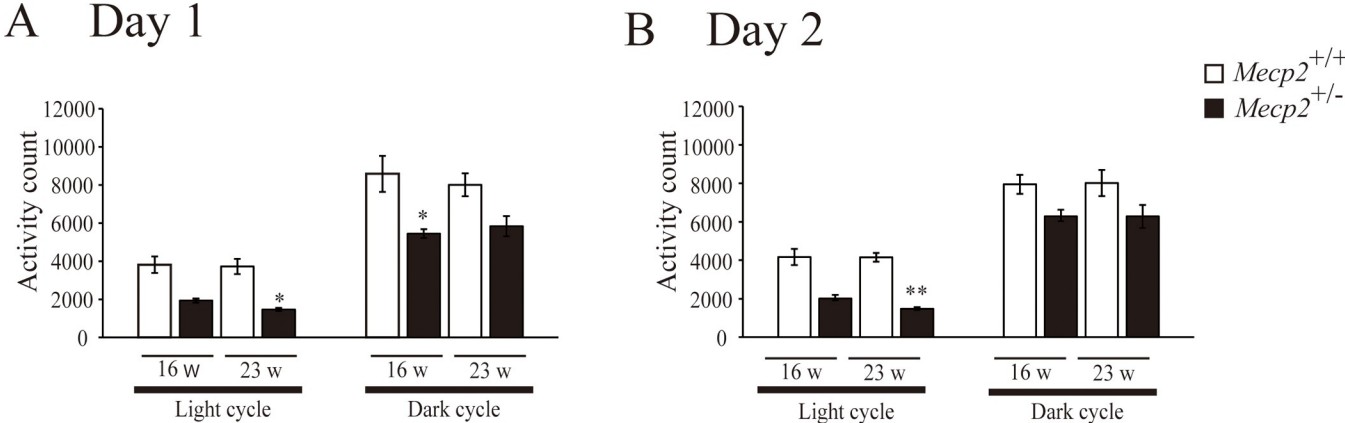

**Fig 4. Impaired locomotor activity among female *Mecp2*$^{+/-}$ rats during both light and dark phases at 16 and 23 weeks of age.** Bar graphs are mean ± SE (n = 6 rats per genotype). $^{*}$p < 0.05, and $^{**}$p < 0.01 vs. age-matched female *Mecp2*$^{+/+}$ rats (Kruskal–Wallis with Dunn's *post hoc* multiple comparisons test).

time and speed at which the rat fell off were lower in the *Mecp2*$^{+/-}$ group compared with the *Mecp2*$^{+/+}$ group (Fig 5A and 5B). Thus, based on the RTT symptoms, both spontaneous and task-evoked motor activity were impaired by *Mecp2*$^{+/-}$ deficiency.

### Impaired spatial learning and memory in *Mecp2*-deficient female rats

To examine the effect of *Mecp2* deficiency on spatial learning and memory, we compared escape latency and target memory between female *Mecp2*$^{+/-}$ and *Mecp2*$^{+/+}$ rats in the MWM starting on 27 w (first acquisition trial). Compared to *Mecp2*$^{+/+}$ rats, *Mecp2*$^{+/-}$ rats

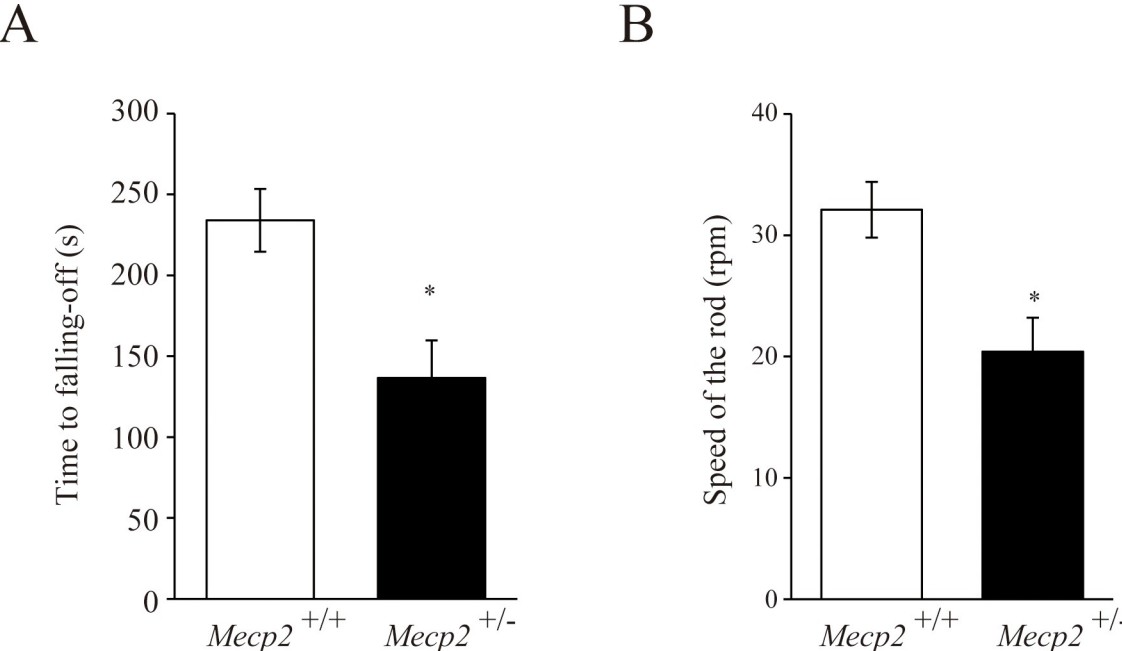

**Fig 5. Impaired motor coordination among female *Mecp2*$^{+/-}$ rats at 26 weeks of age.** The animal was placed on a rotarod, which was accelerated from 4 to 40 rpm. The time at which the animal fell off (A) and the speed at which the animal fell off (B) were measured. Bar graphs are mean ± SE (n = 6 rats per genotype). $^{*}$p < 0.05 vs. female *Mecp2*$^{+/+}$ rats (unpaired two-tailed Mann–Whitney U-test).

demonstrated significantly longer escape latencies (Fig 6B) and swimming distances (Fig 6C) over the four acquisition trial days. Further, *Mecp2*$^{+/-}$ rats made fewer crossing over the former platform location (Fig 6D) and tended to spend less time in the target (former platform) quadrant (Fig 6E). Thus, *Mecp2* deficiency appears to impair spatial learning and memory.

## Reduced acetylcholine but relatively normal biogenic amine levels in multiple brain regions of *Mecp2*-deficient rats

Several studies have reported significant downregulation of biogenic amines in the brains of *Mecp*2 mouse mutants [27–31]. Surprisingly, however, biogenic amine and its metabolite levels (Table 1) and the glutamate/GABA ratio (Table 2) were also near normal in most measured regions of female *Mecp2*$^{+/-}$ rats. The genotypic differences in the biogenic amine/metabolite levels were observed in such brain regions as follows: DA and DOPAC levels were comparable to those in the caudate nucleus and medulla oblongata, although HVA level tended to be lower in the caudate nucleus and was slightly but significantly lower in the medulla oblongata of female *Mecp2*$^{+/-}$ rats. 5-HT level was lower in the thalamus and hypothalamus of female *Mecp2*$^{+/-}$ rats (Table 1). By contrast, the relative reductions in acetylcholine level were greater than those of other transmitters, including biogenic amines: acetylcholine level was substantially lower in the amygdala, hippocampus, caudate nucleus, and medulla oblongata (Table 3). These results suggest that impaired cholinergic signaling could severely affect the behavioral abnormalities observed in *Mecp2*$^{+/-}$ rats. Indeed, cholinergic inputs to the hippocampus are essential for spatial learning and memory [32, 33].

## Changes in GFAP-positive astrocytes and serum IGF

To further examine neurodevelopment abnormalities associated with *Mecp2* deficiency, we examined the number and morphological characteristics of GFAP-positive astrocytes. Astrocytes are known to support neuronal function through production and secretion of various extrinsic factors and by the uptake of neurotransmitters, and the number and morphological features are associated with neurodevelopmental abnormalities as well as neurodegeneration and neuroinflammation [34, 35]. Further, abnormal astrocyte morphology has been observed in RTT patients and models [36, 37]. Comparing with those of female *Mecp2*$^{+/+}$ rats (hippocampus, Fig 7A, a-c; frontal cortex, Fig 7B, a and b; amygdala, Fig 8A, a and b; hypothalamus, Fig 8B, a and b; respectively), the somata of GFAP-positive cells were smaller and possessed less complex processes in the hippocampus (Fig 7A, d-f), frontal cortex (Fig 7B, c and d) and amygdala (Fig 8A, c and d) but not in the hypothalamus (Fig 8B, c and d) of female *Mecp2*$^{+/-}$ rats. In addition, cell numbers were significantly lower in frontal cortex, hippocampus and amygdala, but not in hypothalamus (Table 4). Finally, we also examined potential neurotrophin signaling deficits by measuring serum levels of IGF and BDNF [38]. Both neurotrophic factors play important roles in survival, differentiation and functional development for brain neuronal circuits. Compared to female *Mecp2*$^{+/+}$ rats, plasma IGF-I was significantly reduced but plasma BDNF was normal in the female *Mecp2*$^{+/-}$ rats (Table 5).

## Discussion

We demonstrated that *Mecp2*-deficient female rats (*Mecp2*$^{+/-}$) exhibit multiple behavioral abnormalities resembling those of RTT, including progressively impaired social behavior with reduced self-grooming (at 8, 12, and 23 w, Figs 2 and 3), reduced spontaneous locomotor activity (at both 16 and 23 w, Fig 4, and S1 Fig), poor motor coordination (at 26 w, Fig 5), and deficient spatial cognition (at 27 w, Fig 6). Further, these animals exhibited substantially lower acetylcholine levels in multiple brain regions compared to wild types (*Mecp2*$^{+/+}$), suggesting

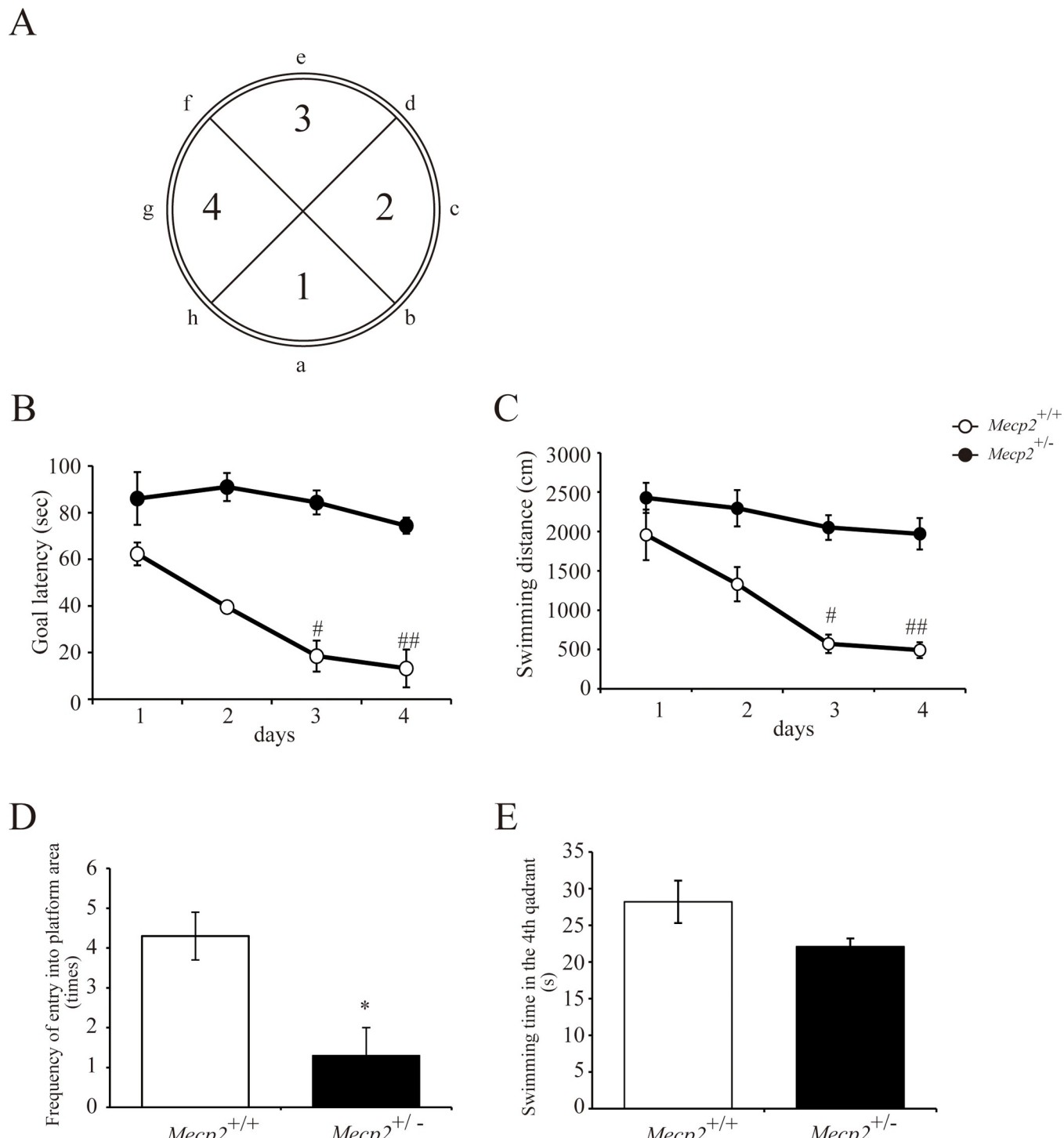

**Fig 6. Impaired spatial learning and memory among female *Mecp2*[+/−] rats in the Morris water maze test.** The Morris water maze test was performed at 27 weeks of age. (A) Images of the Morris water maze indicating the release positions (a–h) and the first to fourth (target) quadrants. Acquisition trials for spatial learning with a hidden escape platform in quadrant 4 were performed twice per day for 4 days (8 times in total). Escape latency (B) and swimming distance (C) were measured. Graphs are mean ± standard error of the mean (SE) for the two daily trials (n = 6 rats per genotype). [#]p < 0.05 and [##]p < 0.01 vs. female *Mecp2*[+/+] rats on the same acquisition day, Friedman test). A probe trial was performed without the hidden platform for the confirmation of spatial memory on the day after completing acquisition trials. The frequency of entry into the platform area (D) and swimming time in the fourth (target) quadrant (E) were measured. Bar graphs are mean ± SE (n = 6 rats per genotype). *p < 0.05 vs. female *Mecp2*[+/+] rats (unpaired two-tailed Mann–Whitney U-test).

**Table 1. Monoamine and monoamine metabolite levels in various brain regions of wild-type (*Mecp2*$^{+/+}$) and Mecp2-deficient (*Mecp2*$^{+/-}$) female rats.**

| | | Levels (ng/g wet weight) | | | | | | |
|---|---|---|---|---|---|---|---|---|
| | | NE | MHPG | DA | DOPAC | HVA | 5-HT | 5-HIAA |
| **Frontal cortex** | *Mecp2*$^{+/+}$ | 521.4 ± 69.2 | 281.9 ± 38.0 | 67.0 ± 7.5 | 23.3 ± 2.0 | 6.9 ± 0.9 | 193.8 ± 27.2 | 130.9 ± 16.3 |
| | *Mecp2*$^{+/-}$ | 400.2 ± 16.5 | 321.5 ± 34.5 | 83.2 ± 4.3 | 30.6 ± 2.7 | 6.7 ± 1.6 | 215.7 ± 23.6 | 162.0 ± 10.7 |
| **Amygdala** | *Mecp2*$^{+/+}$ | 645.7 ± 55.9 | 239.4 ± 24.3 | 132.4 ± 14.8 | 23.1 ± 5.1 | 8.9 ± 2.7 | 263.9 ± 44.3 | 170.1 ± 22.7 |
| | *Mecp2*$^{+/-}$ | 590.7 ± 79.7 | 313.2 ± 45.8 | 103.1 ± 22.8 | 19.3 ± 2.1 | 5.8 ± 1.1 | 231.6 ± 61.3 | 190.9 ± 33.4 |
| **Hippocampus** | *Mecp2*$^{+/+}$ | 741.6 ± 59.7 | 334.4 ± 17.9 | 23.0 ± 2.1 | 1.4 ± 0.4 | 1.1 ± 0.1 | 200.1 ± 15.9 | 243.5 ± 10.9 |
| | *Mecp2*$^{+/-}$ | 606.7 ± 43.0 | 368.2 ± 30.9 | 23.8 ± 1.7 | 1.6 ± 0.3 | 0.9 ± 0.0 | 175.5 ± 8.2 | 220.8 ± 9.1 |
| **Caudate nucleus** | *Mecp2*$^{+/+}$ | 21.2 ± 0.9 | 194.8 ± 17.9 | 4106.8 ± 267.1 | 621.4 ± 62.6 | 183.2 ± 28.0 | 92.9 ± 8.0 | 207.5 ± 16.2 |
| | *Mecp2*$^{+/-}$ | 31.0 ± 5.2 | 263.4 ± 51.5 | 4188.6 ± 472.2 | 607.6 ± 82.7 | 125.4 ± 28.8 | 89.5 ± 14.8 | 216.0 ± 38.2 |
| **Thalamus** | *Mecp2*$^{+/+}$ | 858.0 ± 32.8 | 324.6 ± 21.8 | 131.3 ± 3.3 | 16.7 ± 1.3 | 9.4 ± 1.6 | 512.0 ± 11.0 | 447.5 ± 18.0 |
| | *Mecp2*$^{+/-}$ | 847.9 ± 56.7 | 349.0 ± 26.5 | 114.7 ± 10.4 | 15.8 ± 1.9 | 7.3 ± 1.2 | 423.4 ± 22.5* | 422.8 ± 27.9 |
| **Hypothalamus** | *Mecp2*$^{+/+}$ | 2244.2 ± 160.9 | 286.9 ± 30.2 | 234.1 ± 44.0 | 30.4 ± 5.3 | 5.3 ± 0.9 | 439.3 ± 29.6 | 311.5 ± 27.7 |
| | *Mecp2*$^{+/-}$ | 2150.7 ± 286.9 | 328.5 ± 30.9 | 192.5 ± 20.7 | 31.4 ± 5.6 | 5.2 ± 1.1 | 322.2 ± 38.7* | 266.7 ± 19.5 |
| **Medulla oblongata** | *Mecp2*$^{+/+}$ | 940.1 ± 37.6 | 435.0 ± 34.1 | 39.8 ± 1.4 | 7.4 ± 0.6 | 5.6 ± 0.9 | 432.5 ± 14.2 | 219.9 ± 11.3 |
| | *Mecp2*$^{+/-}$ | 1004.4 ± 35.3 | 540.3 ± 37.6 | 38.5 ± 0.9 | 6.4 ± 0.4 | 3.2 ± 0.2* | 434.3 ± 22.6 | 225.9 ± 12.7 |

NE: norepinephrine; MHPG: 3-methoxyphenyl-4-hydroxy-glycol hemipiperazium; DA: dopamine; DOPAC: 3,4-dihydroxyphenylacetic acid; HVA: homovanillic acid;

5-HT: 5-hydroxytryptamine (serotonin); 5-HIAA: 5-hydroxy indole-3-acetic acid.

Data was expressed as mean ± SE (n = 6 samples per region and genotype).

*p < 0.05 female *Mecp2*$^{+/-}$ vs. female *Mecp2*$^{+/+}$ rats by unpaired two-tailed Mann–Whitney U-test.

pervasive deficits in cholinergic signaling. The phenotype of this rat model basically recapitulates that of *Mecp2*-deficient mouse mutants [8, 17, 18] but in addition also suggests the importance of cholinergic signaling deficits to RTT pathogenesis or symptom expression. In fact, the regional reductions in acetylcholine were relatively larger than the reductions or elevations in other neurotransmitters and their metabolites, including the changes in monoamine levels. The relative stability of monoamine levels is in contrast to predominantly male *Mecp2*$^{tm\ 1.1\ Bird}$ exon 3/4 null mutant mice [27–30] and *Mecp2*$^{tm\ 1.1\ Jae}$ exon 3 null mutant mice [31]. We suggest that these low acetylcholine levels across multiple brain regions may contribute to the postnatal behavioral abnormalities observed in *Mecp2*$^{+/-}$ rats and possibly to RTT [4, 5].

## Progressive impairments in social behavior and abnormalities in motor coordination and spatial cognition

Patients with RTT have autistic symptoms such as social behavior abnormalities. While these symptoms have been reported to improve with age [39], some persist through life [40, 41]. Consistent with findings in female *Mecp2*$^{tm1.1Bird}$ mice [42], female *Mecp2*$^{+/-}$ rats demonstrated a lower frequency of social contact in the open field, and both frequency and duration of contact decreased with age (from 8 to 23 w). Certain abnormalities in social behavior have been observed among *Mecp2*$^{+/-}$ female rats even before 4 w [17, 18], so social deficits appear

**Table 2. Glutamate/GABA ratios across brain regions.**

| | Frontal cortex | Amygdala | Hippocampus | Caudate nucleus | Thalamus | Hypothalamus | Medulla oblongata |
|---|---|---|---|---|---|---|---|
| *Mecp2*$^{+/+}$ | 7.9 ± 0.9 | 5.5 ± 0.3 | 6.5 ± 0.3 | 6.9 ± 0.4 | 2.3 ± 0.2 | 1.4 ± 0.2 | 4.7 ± 0.3 |
| *Mecp2*$^{+/-}$ | 8.3 ± 0.8 | 6.1 ± 0.8 | 6.9 ± 0.4 | 5.0 ± 0.5 | 2.5 ± 0.1 | 1.5 ± 0.1 | 5.1 ± 0.6 |

Data expressed as mean ± SE (n = 6 for each genotype). Non significance female Mecp2$^{+/-}$ vs. Mecp2$^{+/+}$ rats (unpaired two-tailed Mann–Whitney U-test).

**Table 3. Acetylcholine levels across brain regions.**

|  | Frontal cortex | Amygdala | Hippocampus | Caudate nucleus | Thalamus | Hypothalamus | Medulla oblongata |
|---|---|---|---|---|---|---|---|
| $Mecp2^{+/+}$ | 480.9 ± 129.6 | 1066.1 ± 139.1 | 949.0 ± 86.3 | 742.1 ± 77.4 | 771.5 ± 49.0 | 546.7 ± 80.8 | 776.1 ± 146.0 |
| $Mecp2^{+/-}$ | 333.1 ± 38.5 | 475.7 ± 85.5** | 649.4 ± 64.6* | 478.3 ± 35.6* | 675.8 ± 82.8 | 376.4 ± 81.9 | 401.8 ± 23.0** |

Data expressed as mean ± SE (n = 6 for each genotype).

*p < 0.05 and

**p < 0.01 female Mecp2+/- vs. Mecp2+/+ rats (unpaired two-tailed Mann–Whitney U-test).

to be among the earliest onset and most severely progressive. However, the influence of progressive motor dysfunction on social behaviors cannot be excluded, especially at early ages [17, 18] (Figs 2 and 3). For instance, Adcock et al. reported no significant differences in social behavior between female $Mecp2^{+/-}$ and $Mecp2^{+/+}$ rats following auditory discrimination training or skilled forelimb motor training, suggesting that poor perceptual and (or) motor skills may contribute to social deficits in untrained rats [43]. Further, environmental enrichment [44, 45], which is known to improve motor function and cognition, can partially mitigate the developmental symptoms in female $Mecp2^{+/-}$ rats. Despite such limitations, the influence of training for behavioral test battery and environment and so on, should be considered on the interpretation of results; the consistency and progressive nature of these social deficits may be useful for preclinical studies on treatments to improve autistic symptoms among RTT patients.

Poor spatial memory is also a critical feature of the rat model as severe intellectual disability is a hallmark of RTT [46, 47]. Although spatial learning and memory impairments are not included in the diagnostic criteria for RTT [3, 48, 49], spatial learning and memory are critical for rodent survival and thus impairments are sensitive indicators of brain maldevelopment and dysfunction, particularly hippocampal dysfunction. Further, hippocampal dysfunction is a shared cause of cognitive deficits among numerous neurodegenerative and neurodevelopmental diseases. In this study, we used MWM [50, 51] to demonstrate that spatial learning and memory are markedly impaired in adult (29W) female $Mecp2^{+/-}$ rats (Fig 6). The effects of $Mecp2$ gene mutations on spatial learning in the MWM are known to vary by mutation type, with impairments observed among $Mecp2^{tm\ 1.1\ Bird}$ mice [43, 44] but not female $Mecp2^{\ tm\ 1.1\ Jae}$ mice [52, 53]. Alternatively, abnormalities in recognition memory have been observed across many mutations. Since the $Mecp2^{+/-}$ rat is an outbred strain, it may be useful for extracting phenotypes common across species and genotypes. Although further studies using inbred strains and rats with different $Mecp2$ mutations are needed, the differences in hippocampus-dependent cognition between mouse and rat models may provide important clues to the underlying neuroanatomical and neurochemical changes. For instance, our current findings suggest that poor MWM performance may stem from deficient cholinergic signaling in the hippocampus of female $Mecp2^{+/-}$ rats.

## Relationships between cholinergic and behavioral abnormalities

Many neuroanatomical and neurochemical changes have been reported in RTT patients, but the molecular mechanisms through which $MECP2$ gene mutations lead to pathological alterations are unknown. Acetylcholine level was significantly reduced in multiple brain regions of female $Mecp2$-deficient rats and associated with impaired social behavior, motor activity, and spatial cognition (Table 3). In particular, the importance of cholinergic neurons to cognitive function and mood disorders is well established [32, 33]. Clinical studies have shown decreases in the cholinergic neuron number [54], acetylcholine biosynthetic enzymes choline

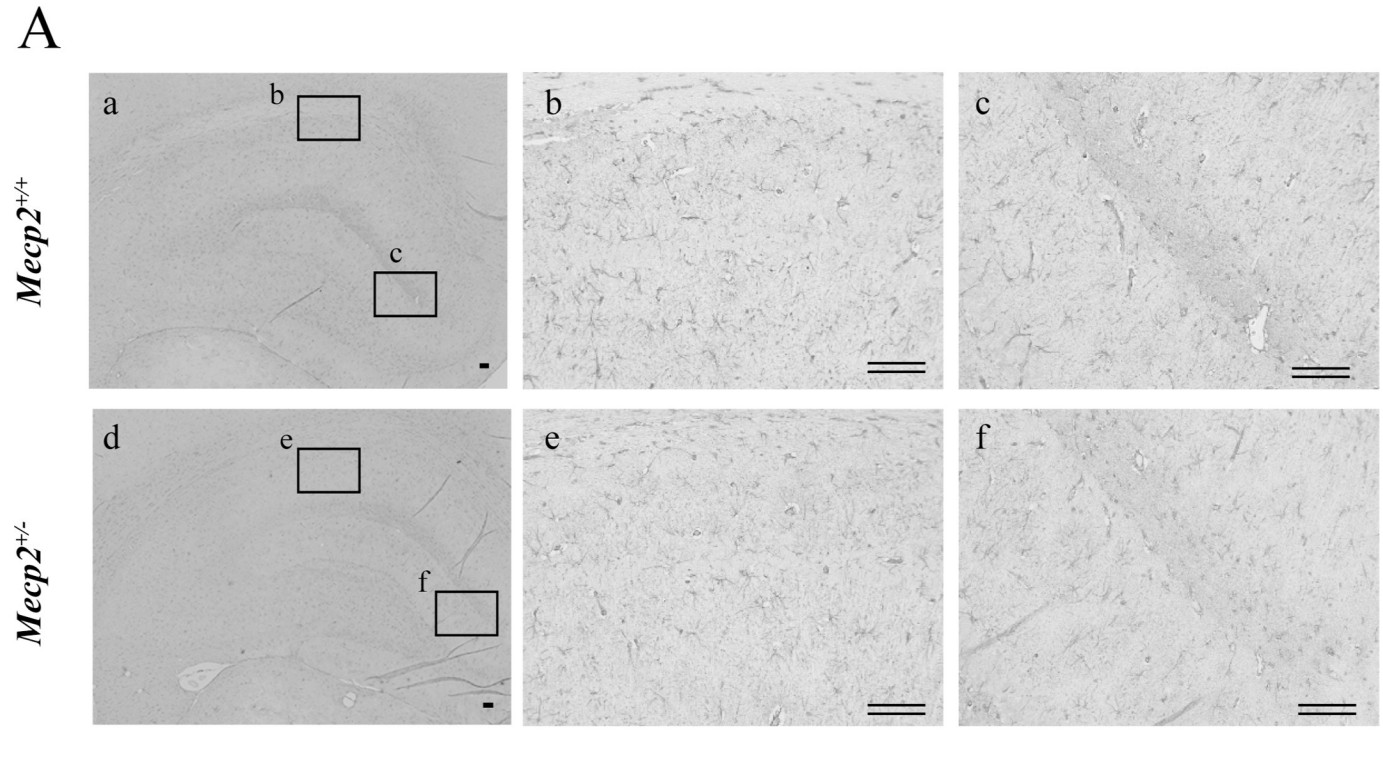

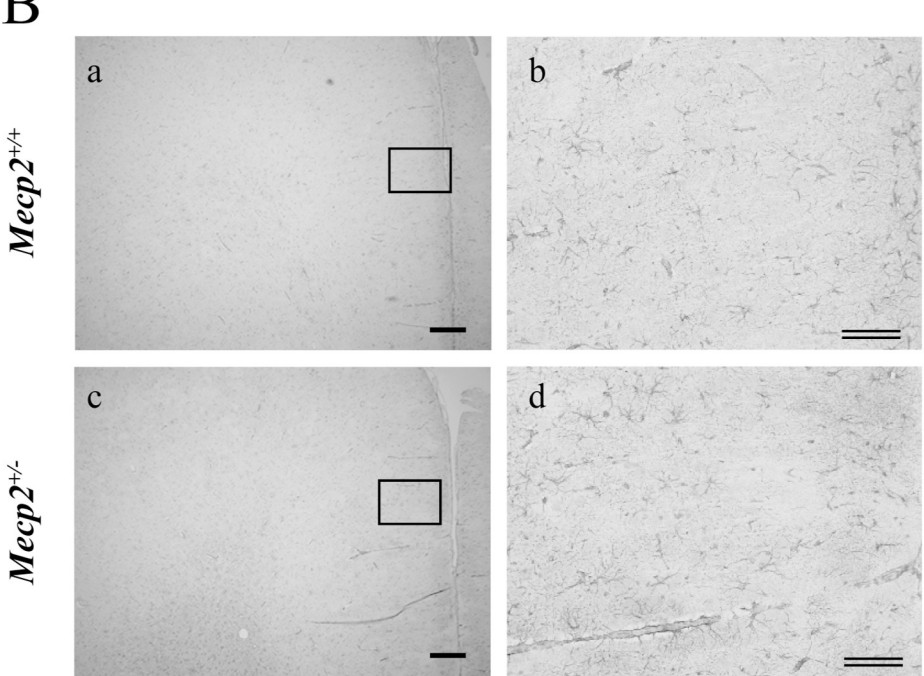

**Fig 7. Changes in the number of glial fibrillary acidic protein (GFAP)-positive cells (astrocytes) in the hippocampus and frontal cortex of a female *Mecp2*[+/−] rat brain.** The brain was collected one week after the Morris water maze test and immunostained for the astrocytic marker GFAP. Shown are the morphology and the number of GFAP-positive cells in the hippocampus (A: a-c: *Mecp2*[+/+], b and c: higher magnification of square in a; d-f, *Mecp2*[+/−], e and f: higher magnification of square in d), the frontal cortex (B: a and b: *Mecp2*[+/+]; c, d: *Mecp2*[+/−]; b and d: higher magnification of square in a and c, respectively). Scale bar = 300 μm; double bars = 100 μm, respectively.

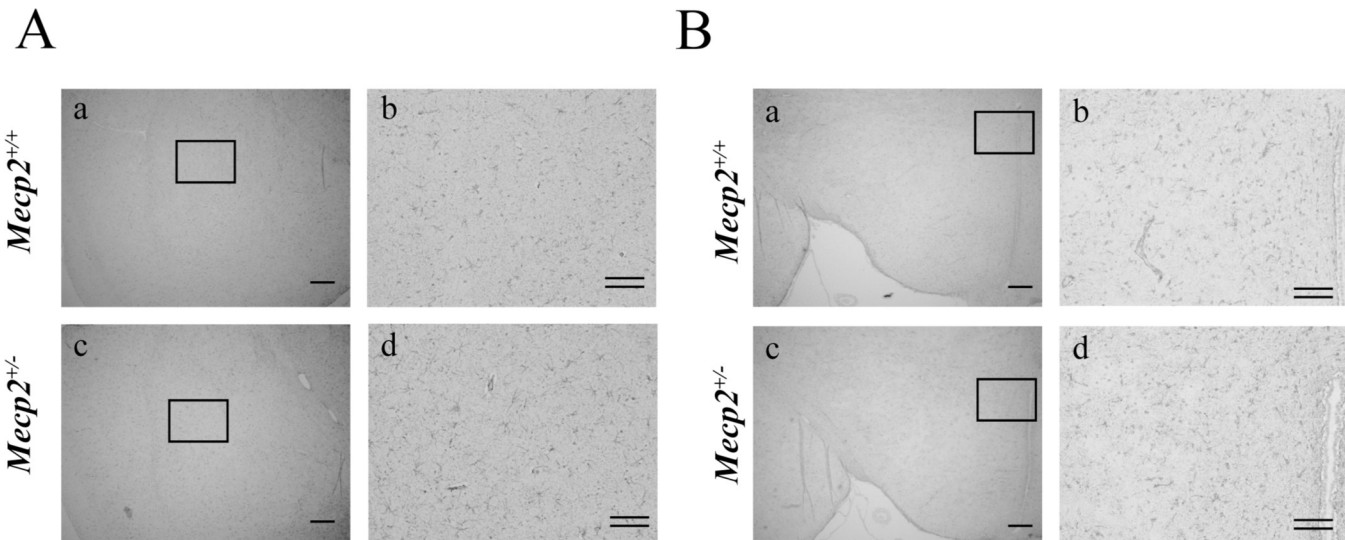

**Fig 8. Number of glial fibrillary acidic protein (GFAP)-positive cells (astrocytes) in the hypothalamus and amygdala of a female *Mecp2*$^{+/-}$ rat brain.**
Shown are the morphology and number of GFAP-positive cells (A. amygdala, B. hypothalamus: a and b: *Mecp2*$^{+/+}$; c and d: *Mecp2*$^{+/-}$; b and d: higher magnification of square in a and c, respectively). Scale bar = 300 μm; double bars = 100 μm, respectively.

acetyltransferase (ChAT) and vesicular acetylcholine transporter activities, and cholinergic receptor expression [55, 56] in the brains of patients with RTT, and the changes in cholinergic neuron markers are linked to the severity of symptoms [57, 58]. Abnormalities in the cholinergic system have also been reported in *Mecp2* mutant mice, including reduced ChAT expression in basal forebrain and striatum [59, 60] and locus coeruleus (LC) neurons [61]. In addition, abnormalities in memory performance and social behavior among *Mecp2* null mutants were reversed by administration of acetylcholine receptor agonists and the cholinesterase inhibitor donepezil [58, 62].

While acetylcholine level was markedly reduced in female *Mecp2*$^{+/-}$ rats, relative changes in monoamines, monoamine metabolites, and the glutamic acid/GABA ratio were smaller and observed in a more limited number of brain regions. This result was unexpected because the levels of many monoamine metabolites as well as expression levels of rate-limiting biosynthetic enzymes have been shown to decline with age in male *Mecp2*$^{tm1.1Bird}$ mice [27, 30]. The result was also unexpected because the disorders of involuntary movements in RTT (progressive rigidity, dyskinesia, and dystonia) have been suggested to be associated with the maldevelopment of the dopaminergic nervous system [63–65]. It is possible that some compensatory mechanism may make it difficult to detect impairment in the monoaminergic nervous system

**Table 4. Number of GFAP-positive cells across brain regions.**

|  | Number of GFAP-positive cells/slice | |
| --- | --- | --- |
|  | *Mecp2*$^{+/+}$ | *Mecp2*$^{+/-}$ |
| **Frontal cortex** | 1108.2 ± 33.5 | 609.5 ± 66.0 ** |
| **Hippocampus** | 476.5 ± 15.4 | 347.3 ± 11.7 ** |
| **Amygdala** | 51.8 ± 1.5 | 49.5 ± 2.3** |
| **Hypothalamus** | 39.8 ± 4.6 | 39.0 ± 2.7 |

Data expressed as mean ± SE (n = 6 for each genotype).

$^{**}$p < 0.01 female Mecp2$^{+/-}$ vs. Mecp2$^{+/+}$ rats (unpaired two-tailed Mann–Whitney U-test).

**Table 5. Plasma IGF and BDNF levels.**

|  | *Mecp2*$^{+/+}$ | *Mecp2*$^{+/-}$ |
|---|---|---|
| **IGF-1 (ng/mL)** | 1080.2 ± 70.3 | 769.4 ± 52.4 ** |
| **BDNF (pg/mL)** | 46.4 ± 5.9 | 51.9 ± 18.2 |

Data expressed as mean ± SE [IGF-1: n = 6 and BDNF: n = 2 (n = 4, under detection limit), for each genotype, respectively)].

**p < 0.01 female Mecp2$^{+/-}$ vs. Mecp2$^{+/+}$ rats (unpaired two-tailed Mann–Whitney U-test).

because approximately half of the neurons in female *Mecp2*-mutant mouse brain maintain normal *Mecp2* expression [31, 36]. Recently, Wong et al. reported that the density of dopamine D2 receptor was significantly decreased in the striatum of *Mecp2*$^{tm\ 1.1\ Bird}$ null/heterozygous mutant mice at 7–10 w but not after 15 w. They pointed out that such developmental stage-dependent alteration could affect the measured density of D2 receptor in the RTT female brain postmortem [66]. Therefore, we could not exclude the possibility that some behavioral abnormalities observed in this study may be influenced by monoaminergic disturbances—particularly at earlier developmental stages. In fact, 5-HT levels in the thalamus and hypothalamus as well as the level of the dopamine metabolite HVA in the medulla oblongata were significantly reduced (Table 1), suggesting some abnormalities in the monoaminergic system development.

Self-grooming behavior in *Mecp2*$^{+/-}$ rats was reported to be equivalent to that of the control group (at 4 w [18]) and tended to be, or even was, significantly decreased in this study (at 8 w, 12 w, and 23 w, Fig 3). Excessive self-grooming behavior is known to be suppressed by acetylcholine antagonists [67–69]. Therefore, progressive impairment in the cholinergic nervous system is likely to explain the self-grooming behavioral abnormalities. Excessive self-grooming behavior, however, is a typical symptom of autism model mice [21–23] and has also been observed in *Mecp2* null male mice [24–26]. Monoamine reductions have been found not only in postmortem brain tissue but also in the cerebrospinal fluid (CSF) of patients with RTT [28, 70]; CSF measures during development should therefore be performed in the rat model also to clarify the relationship of disturbances in the cholinergic/monoaminergic system and behavioral changes. Such efforts, in future studies, would lead to an understanding of the complex pathological changes due to mutation in the *MECP2* gene, leading to the development of optimal drug treatment strategies.

## Value of the *Mecp2*$^{+/-}$ rat model for RTT diagnosis and treatment

We also observed reduced numbers of astrocytes (GFAP-positive cells) in the frontal cortex and hippocampus and as well as lower plasma IGF in female *Mecp2*$^{+/-}$ rats (Fig 7 and Tables 4 and 5), consistent with previous findings of stunted morphological and functional development of astrocytes [36, 37, 71] and reduced IGF signaling [38, 72, 73] in both RTT patients and *Mecp2* mutant mice. Astrocytes regulate neuronal functions through production and secretion of growth factors, nutrients, and cytokines and by the uptake of transmitters and metabolites [74, 75], while IGF is involved in the differentiation and functional maintenance of neurons as a blood–brain barrier permeable neurotrophic factor [76, 77]. The identification of blood or CSF biomarkers associated with specific neuropathological processes may aid in RTT diagnosis and provide clues to pathogenic mechanisms, and the female *Mecp2*$^{+/-}$ rat model may be a powerful tool for this purpose. Candidate biomarkers for pathogenesis may include metabolites generated by astrocytes [78] and/or molecules associated with IGF signaling [79].

## Supporting information

**S1 Fig. Impaired locomotor activity among female *Mecp2*$^{+/−}$ rats during both light and dark phases at 16 and 23 weeks of age.** A-D: 24-h locomotor activity in female *Mecp2*$^{+/+}$ (open symbols) and *Mecp2*$^{+/-}$ (filled symbols) rats. Data are the average ± SE of data collected at each half an hour from 16 w (A, B) or 23 w (C, D) old rats over 2 consecutive 24-h periods (A, C: day1; B, D: day2) (n = 6, for each geneotype). Non significance female Mecp2$^{+/-}$ vs. Mecp2$^{+/+}$ rats (Kruskal–Wallis with Dunn's *post hoc* multiple comparisons test).
(TIF)

## Acknowledgments

We would like to thank Editage (www.editage.jp) for English language editing.

## Author Contributions

**Data curation:** Hiroyasu Murasawa.

**Formal analysis:** Hiroyasu Murasawa, Hitomi Soumiya.

**Investigation:** Hiroyasu Murasawa, Hiroyuki Kobayashi, Jun Imai.

**Methodology:** Hiroyasu Murasawa, Takahiko Nagase.

**Project administration:** Takahiko Nagase.

**Supervision:** Takahiko Nagase.

**Writing – original draft:** Hiroyasu Murasawa, Hitomi Soumiya, Hidefumi Fukumitsu.

**Writing – review & editing:** Hidefumi Fukumitsu.

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
