## [Decision Letter · Decision Letter 0]

20 Apr 2021

PONE-D-21-05550

Substantial acetylcholine reduction in multiple brain regions of Mecp2-deficient female rats and associated behavioral abnormalities

PLOS ONE

Dear Dr. Fukumitsu,

Thank you for submitting your manuscript to PLOS ONE. After careful consideration, we feel that it has merit but does not fully meet PLOS ONE’s publication criteria as it currently stands. Therefore, we invite you to submit a revised version of the manuscript that addresses the points raised during the review process.

We look forward to receiving your revised manuscript.

Kind regards,

Atsushi Asakura, Ph.D

Academic Editor

PLOS ONE

Journal Requirements:

2. In line with PLOS' guidelines on the reporting of animal research (https://journals.plos.org/plosone/s/submission-guidelines#loc-animal-research), please include in your Methods section the total number of animals used in the study.

As part of your revision, please complete and submit a copy of the ARRIVE Guidelines checklist, a document that aims to improve experimental reporting and reproducibility of animal studies for purposes of post-publication data analysis and reproducibility: https://arriveguidelines.org/sites/arrive/files/Author%20Checklist%20-%20Full.pdf. Please include your completed checklist as a Supporting Information file.

Note that if your paper is accepted for publication, this checklist will be published as part of your article.

'The funders had no role in study design, data collection and analysis, decision to publish, or preparation of the manuscript.'

Reviewers' comments:

Reviewer's Responses to Questions

**Comments to the Author**

1. Is the manuscript technically sound, and do the data support the conclusions?

Reviewer #1: Yes

Reviewer #2: Yes

2. Has the statistical analysis been performed appropriately and rigorously? 

Reviewer #1: No

Reviewer #2: Yes

3. Have the authors made all data underlying the findings in their manuscript fully available?

Reviewer #1: Yes

Reviewer #2: No

4. Is the manuscript presented in an intelligible fashion and written in standard English?

Reviewer #1: Yes

Reviewer #2: Yes

5. Review Comments to the Author

Reviewer #1: Review

March 19, 2021

PONE-D-21-05550

Substantial acetylcholine reduction in multiple brain regions of Mecp2-deficient female rats and associated behavioral abnormalities

The authors sought to assess the effects of Mecp2 gene deletion on female rats (Mecp2+/−) by examining social behavior, motor function, spatial cognition and other physiological parameters. The study was well-designed and well-constructed. This study represents major advances in the study of animal models of Rett syndrome. The manuscript will likely merit publication with revisions.

Financial Disclosure

Please include information about the funding for this study.

Include a Financial Disclosure.

Enter a financial disclosure statement that describes the sources of funding for the work included in this submission.

Materials and methods

Morris water maze test

Line 127

Since the second figure to be mentioned in the text occurs in this line, the figure should be numbered figure 2.

Statistical Analysis

Are the data normally distributed as demonstrated by the Kolmogorov-Smirnov or Wilks-Shapiro procedures? If the data are not normally distributed, then a nonparametric procedure such as Kruskal-Wallis is appropriate.

Results

Impaired social behaviors in Mecp2-deficient female rats

Page 9

Line 192

Change “disorders” to “disorder”

Table 1

Pages 13 to 15

Express the values of each row in a single line by using a smaller font.

Legend to figure 7

Line 316

The scale bar appears to be the same length in all panels. What is the magnification indicated by the two scale bars?

Relationships between cholinergic and behavioral abnormalities

Line 387-408

Include a discussion of the dopaminergic deficits observed as follows:

Wong DF, Blue ME, Brašić JR,, Nandi A, Valentine H, Stansfield KH, Rousset O, Bibat G, Yablonski ME, Johnston MV, Gjedde A, Naidu SB. Are dopamine receptor and transporter changes in Rett syndrome reflected in Mecp2-deficient mice? Exp Neurol. 2018; 307: 74-81. https://doi.org/10.1016/ j.expneurol.2018.05.019 PubMed PMID: 29782864.

Reference

Line 517

Remove the space between “30.” and “Panayotis”

Lines 549-550

Reference 41.

Include the original title of the article in German as follows:

Rett A. Uber ein bisher nicht bekanntes Krankheitsbild einer angeborenen Stoffwechselst örung [On an until now unknown disease of a congenital metabolic disorder]. Krankenschwester. 1966 Sep;19(9):121-2. German. PMID: 5179620.

Please add appendix of abbreviations.

Define all acronyms before their first use.

Supplemental Methods

The section in the supplemental methods belong in the body of the text.

There are no restrictions on word counts, number of figures, or amount of supporting information.

Reviewer #2: In the manuscript “Substantial acetylcholine reduction in multiple brain regions of Mecp2-deficient female rats and associated behavioral abnormalities” The authors assessed the effects of Mecp2 gene deletion on female rats (Mecp2 +/− ) and found severe impairments in social behavior, motor function, and spatial cognition and lower plasma IGF-1 (but not BDNF) and markedly reduced acetylcholine (30%–50%) in multiple brain regions compared to female Mecp2 +/+ rats.

This is a purely descriptive study which could be informative but some of the results appears to be disconnected. The correlation of behavioral changes with acetylcholine levels is appears far reaching based on the way the manuscript is organized.

Figure 7 images are of poor quality and images of amygdala and hypothalamus should be included.

Page 2, Line 25. “…..spatial cognition as well lower plasma” - the word “as” is missing.

Page 3, Line 45. “One of most striking findings” - the word “the” is missing

Page 9, Line 188. “….and the both” – remove “the”

Page 9, Line 193. The author report a decrease in repetitive grooming behavior. Veeraragavan et al. 2016 reported no significant changes in Mecp2-/+ mice, what do you attribute to the divergent results?

Page 22, Line 403: “….GABAA (GABAnergic….” – should be “GABAergic”

Figure 4 A and B. For consistency capitalize the first word on the y axis.

Figure 6 B and C. For consistency capitalize the words on the y axis “goal latency” and “swimming distance”.

6. PLOS authors have the option to publish the peer review history of their article (what does this mean?). If published, this will include your full peer review and any attached files.

Reviewer #1: **Yes: **James Brasic

Reviewer #2: No

---

## [Author Response · Author response to Decision Letter 0]

16 Jun 2021

Response to Reviewers:

We wish to express our appreciation to the Reviewers for their insightful comments, which have helped us improve our paper substantially.

Reply to Reviewer #1

Comments to authors

Point 1: Materials and methods Morris water maze test Line 127: Since the second figure to be mentioned in the text occurs in this line, the figure should be numbered figure 2. 

Answer 1: Thank you for the correction. We have accordingly altered the order of the figure number.

Point 2: Materials and methods Statistical Analysis: Are the data normally distributed as demonstrated by the Kolmogorov-Smirnov or Wilks-Shapiro procedures? If the data are not normally distributed, then a nonparametric procedure such as Kruskal-Wallis is appropriate. 

Answer 2: Thank you for the comment. Because sample size for each experiment was small, it was not appropriate to evaluate the equal distribution of the sample. We therefore changed all our statistical methods to nonparametric procedures, accordingly.

Point 3: Results Impaired social behaviors in Mecp2-deficient female rats: Page 9 Line 192 Change “disorders” to “disorder”. 

Answer 3: Thank you for the correction. We have accordingly changed “disorders” to “disorder.”

Point 4: Table 1 Pages 13 to 15 Express the values of each row in a single line by using a smaller font.

Answer 4: Thank you for the correction. In accordance, we have corrected the table to express the values of each row in a single line using a smaller font.

Point 5: Legend to figure 7 Line 316 The scale bar appears to be the same length in all panels. What is the magnification indicated by the two scale bars?

Answer 5: Thank you for the correction. In accordance, we have corrected the scale bars.

Point 6: Discussion Relationships between cholinergic and behavioral abnormalities Line 387-408 Include a discussion of the dopaminergic deficits observed as follows: Wong DF, Blue ME, Brašić JR,, Nandi A, Valentine H, Stansfield KH, Rousset O, Bibat G, Yablonski ME, Johnston MV, Gjedde A, Naidu SB. Are dopamine receptor and transporter changes in Rett syndrome reflected in Mecp2-deficient mice? Exp Neurol. 2018; 307: 74-81. https://doi.org/10.1016/ j.expneurol.2018.05.019 PubMed PMID: 29782864.

Answer 6: Thank you for the comment and information. In accordance, we have thoroughly corrected the part of the Discussion section concerning the possible involvement of dopaminergic deficits in the behavioral abnormalities of Mecp2+/− rats and cited the article.

Point 7: Reference Line 517 Remove the space between “30.” and “Panayotis”　

Answer 7: Thank you for the correction. We have accordingly removed the space between “30.” and “Panayotis.”

Point 8: Lines 549-550 Reference 41. Include the original title of the article in German as follows: Rett A. Uber ein bisher nicht bekanntes Krankheitsbild einer angeborenen Stoffwechselst örung [On an until now unknown disease of a congenital metabolic disorder]. Krankenschwester. 1966 Sep;19(9):121-2. German. PMID: 5179620.

Answer 8: Thank you for the correction. In accordance, we have corrected the reference to include the original title of the article in German.

Point 9: Please add appendix of abbreviations. Define all acronyms before their first use.

Answer 9: Thank you for the correction. In accordance, we have added an appendix of abbreviations and defined all acronyms before their first use.

Point 10: Supplemental Methods The section in the supplemental methods belong in the body of the text. There are no restrictions on word counts, number of figures, or amount of supporting information.

Answer 10: Thank you for the comment. In accordance, we have moved the supplemental methods into the body of the text.

 

Reply to reviewer #2

Comments to authors

Point 1: The correlation of behavioral changes with acetylcholine levels is appears far reaching based on the way the manuscript is organized. 

Answer 1: Thank you for the comment. Considering both reviewers’ comments, we have thoroughly corrected the part of the Discussion section concerning the possible involvement of monoaminergic dysfunctions in the behavioral abnormalities of the Mecp2+/− female rats.

Point 2: Figure 7 images are of poor quality and images of amygdala and hypothalamus should be included. 

Answer 2: Thank you for the comment. In accordance, we have replaced the microscopic photographs and also inserted images of the amygdala and hypothalamus in Figure 7 and Figure 8.

Point 3: Page 2, Line 25. “…..spatial cognition as well lower plasma” - the word “as” is missing. 

Answer 3: Thank you for the correction. In accordance, we have added the word “as.”

Point 4: Page 9, Line 193. The author report a decrease in repetitive grooming behavior. Veeraragavan et al. 2016 reported no significant changes in Mecp2-/+ mice, what do you attribute to the divergent results?

Answer 4: Thank you for the comments. Concerning self-grooming, we considered that the divergent results would come from the developmental impairment in the cholinergic system. Excessive self-grooming behavior, however, is a typical symptom of autism model mice and has also been observed in Mecp2 null male mice mutants. Therefore, further investigations are necessary to understand the meaning of this behavioral abnormality in the Mecp2-deficient female rat. We have added these limitations to the Discussion section.

Point 5: Page 22, Line 403: “….GABAA (GABAnergic….” – should be “GABAergic”

Answer 5: Thank you for the correction. In accordance, we have corrected the word to “GABAergic.”

Point 6: Figure 4 A and B. For consistency capitalize the first word on the y axis.

Answer 6: Thank you for the correction. In accordance, we have capitalized the first word on the y-axis. 

Point 7: Figure 6 B and C. For consistency capitalize the words on the y axis “goal latency” and “swimming distance”.

Answer 7: Thank you for the correction. In accordance, we have capitalized the first word on the y-axis.

---

## [Decision Letter · Decision Letter 1]

13 Aug 2021

PONE-D-21-05550R1

Substantial acetylcholine reduction in multiple brain regions of Mecp2-deficient female rats and associated behavioral abnormalities

PLOS ONE

Dear Dr. Fukumitsu,

Thank you for submitting your manuscript to PLOS ONE. After careful consideration, we feel that it has merit but does not fully meet PLOS ONE’s publication criteria as it currently stands. Therefore, we invite you to submit a revised version of the manuscript that addresses the points raised during the review process.

We look forward to receiving your revised manuscript.

Kind regards,

Atsushi Asakura, Ph.D

Academic Editor

PLOS ONE

Journal Requirements:

Reviewers' comments:

Reviewer's Responses to Questions

**Comments to the Author**

1. If the authors have adequately addressed your comments raised in a previous round of review and you feel that this manuscript is now acceptable for publication, you may indicate that here to bypass the “Comments to the Author” section, enter your conflict of interest statement in the “Confidential to Editor” section, and submit your "Accept" recommendation.

Reviewer #1: (No Response)

Reviewer #2: (No Response)

2. Is the manuscript technically sound, and do the data support the conclusions?

Reviewer #1: Yes

Reviewer #2: Yes

3. Has the statistical analysis been performed appropriately and rigorously? 

Reviewer #1: Yes

Reviewer #2: N/A

4. Have the authors made all data underlying the findings in their manuscript fully available?

Reviewer #1: Yes

Reviewer #2: Yes

5. Is the manuscript presented in an intelligible fashion and written in standard English?

Reviewer #1: No

Reviewer #2: Yes

6. Review Comments to the Author

Reviewer #1: The authors compared and contrasted the neurochemical and behavioral aspects of wild type and Mecp2-deficient female rats utilizing a reasonable experimental design and analysis. They concluded that acetylcholine concentration was associated with behavior.

The study was well conducted. The discussion was reasonable.

Some readers may be unfamiliar with some technical details so explicitly documenting them will likely facilitate understanding. Some sections may be better worded and organized to help readers comprehend the key points.

The manuscript will likely merit publication with revisions.

Page 6

Line 131

Change “MWM test” to “Morris water maze (MWM) test”

Page 7

Line 157

Change “Friedman’s test” to “Friedman test”

Provide a reference for the Friedman test. What statistical program was used?

Line 161

Provide a reference for unpaired two-tailed Mann–Whitney U-test. What statistical program was used?

Page 11

Lines 245-246

Provide a reference for unpaired two-tailed Mann–Whitney U-test.

Line 247

Provide a reference for the Kruskal–Wallis with Dunn’s post hoc multiple comparisons test.

Line 248

Provide a reference for the Friedman test.

Page 12

Line 269

Provide a reference for the Friedman test.

Line 270

Provide a reference for the Kruskal–Wallis with Dunn’s post hoc multiple comparisons test.

Line 277

Provide a reference for the Kruskal–Wallis with Dunn’s post hoc multiple comparisons test.

Page 13

Lines 292-293

Provide a reference for the Kruskal–Wallis with Dunn’s post hoc multiple comparisons test.

Lines 298-299

Provide a reference for unpaired two-tailed Mann–Whitney U-test.

Line 308

Change “Fig-2D” to “Fig 2D”

Page 15

Line 327

Change “Mecp2-deficient.” to “Mecp2-deficient”

Page 16

None of the items utilize **.

Line 331

Change “*p < 0.05 and **p < 0.01 vs. female Mecp2+/+” to

“*p < 0.05 female Mecp2+/- vs. Mecp2+/+”

Lines 331-332

Provide a reference for unpaired two-tailed Mann–Whitney U-test.

Page 17

Table 2

None of the items utilize *.

Line 335

Change *p < 0.05 vs. female Mecp2+/+ rats” to

*Non significance female Mecp2+/- vs. Mecp2+/+ rats”

Lines 335-336

Provide a reference for unpaired two-tailed Mann–Whitney U-test.

Line 339

Change “vs. female Mecp2+/+ rats” to “female Mecp2+/- vs. Mecp2+/+ rats”

Provide a reference for unpaired two-tailed Mann–Whitney U-test.

Page 19

Line 361

Change **p <0.01 vs. female Mecp2+/+ rats” to **p <0.01 female Mecp2+/- vs. Mecp2+/+ rats”

Line 362

Provide a reference for unpaired two-tailed Mann–Whitney U-test.

Line 366

Change “**p < 0.01 vs. female Mecp2+/+ rats” to “**p < 0.01 female Mecp2+/- vs. Mecp2+/+ rats”

Page 20

Line 385

Change “demonstrate” to “demonstrated”

Abbreviations

Place the abbreviations in alphabetical order.

Reviewer #2: The authors have addressed the previous comments. Below are some additional minor comments.

Page 4, Line 78. Remove “on so on”

Page 7, Line 150. This is the Figure legend for figure 2, not sure why it is in the methods section.

Page 12, Line 260-261. “Both the frequency (Fig 4A) and duration 261 (Fig. 4B) of self-grooming behavior were significantly or tended to be reduced in female Mecp2+/−…” However, the graph in figure 4B does not show statistical significance for the duration. The current phrasing is not clear, authors should clarify that frequency was reduced and duration tended to be reduced but not significant.

Page 12, Line 283. “Spontaneous locomotor activity was significantly or tended to be lower among female Mecp2+/− rats compared to female Mecp2+/+ rats (Fig 5) during both light and dark phases.” Authors should clarify the specific timepoints of significance.

Page 12, Line 285. For consistency lower case w “29 w”

Page 13, Line 301. It is unclear why the result section “Impaired spatial learning and memory in Mecp2-deficient female rats” is Figure 2, following Figure 6.

Page 14, Line 314. “Surprising, however, biogenic amine and…” – Change to “Surprisingly”

Page 14, Line 323. “These results suggest that impaired cholinergic signaling would severely affect the behavioral abnormalities observed...” The word “would” should be changed to “could”.

Page 18, Line 344. ” Astrocytes are known to support neuronal function through production and secretion of various extrinsic factors and by the uptake of neurotransmitters, and the number and morphological features are associated with neurodevelopmental abnormalities as well as neurodegeneration and neuroinflammation.” A reference is needed for this sentence.

Page 18, Line 355. “Both the neurotrophic factors play important…” – Remove “the”

7. PLOS authors have the option to publish the peer review history of their article (what does this mean?). If published, this will include your full peer review and any attached files.

Reviewer #1: **Yes: **James Robert Brasic

Reviewer #2: No

---

## [Author Response · Author response to Decision Letter 1]

30 Aug 2021

Response to Reviewers:

We wish to express our appreciation to the Reviewers for their insightful comments, which have helped us improve our paper substantially.

Reply to Reviewer #1

Point 1: Page 6 

Line 131 Change “MWM test” to “Morris water maze (MWM) test”. 

Answer 1: Thank you for the correction. We have accordingly spelled out “MWM test”.

Point 2: Page 7 

Line 157 Change “Friedman’s test” to “Friedman test” Provide a reference for the Friedman test. What statistical program was used? 

Line 161 Provide a reference for unpaired two-tailed Mann–Whitney U-test. What statistical program was used?

Answer 2: Thank you for the comment. We have accordingly changed “Friedman’s test” to “Friedman test” Concerning about statistical program, we had used GaphPad Prism (version 9.1.2 for Windows) for all statistical analyses (Friedman test, unpaired two-tailed Mann-Whitney U-test, and Kruskal-Wallis with Dunn’s post hoc multiple comparisons test), according to the software guide. So, we have described it in the revised manuscript. 

Point 3: Page 13 

Line 308 Change “Fig-2D” to “Fig 2D” 

Answer 3: Thank you for the correction. We have accordingly changed “Fig-2D” to “Fig 2D”.

Point 4: Page 15 

Line 327 Change “Mecp2-deficient.” to “Mecp2-deficient

Answer 4: Thank you for the correction. We have accordingly changed “Mecp2-deficient.” to “Mecp2-deficient”.

Point 5: Page 16 

Table 1 None of the items utilize **.

Line 331 Change “*p < 0.05 and **p < 0.01 vs. female Mecp2+/+” to “*p < 0.05 female Mecp2+/- vs. Mecp2+/+”

Answer 5: Thank you for the correction. In accordance, we have changed “*p < 0.05 and **p < 0.01 vs. female Mecp2+/+” to “*p < 0.05 female Mecp2+/- vs. Mecp2+/+”.

Point 6: Page 17 

Table 2 None of the items utilize *.

Line 335 Change *p < 0.05 vs. female Mecp2+/+ rats” to *Non significance female Mecp2+/- vs. Mecp2+/+ rats”.

Line 339 Change “vs. female Mecp2+/+ rats” to “female Mecp2+/- vs. Mecp2+/+ rats”

Answer 6: Thank you for the correction. In accordance, we have changed *p < 0.05 vs. female Mecp2+/+ rats” to *Non significance female Mecp2+/- vs. Mecp2+/+ rats”. We also have changed “vs. female Mecp2+/+ rats” to “female Mecp2+/- vs. Mecp2+/+ rats”.

Point 7: Page 19 

Line 361 Change **p <0.01 vs. female Mecp2+/+ rats” to **p <0.01 female Mecp2+/- vs. Mecp2+/+ rats”

Line 366 Change “**p < 0.01 vs. female Mecp2+/+ rats” to “**p < 0.01 female Mecp2+/- vs. Mecp2+/+ rats”

Answer 7: Thank you for the correction. In accordance, we have changed **p <0.01 vs. female Mecp2+/+ rats” to **p <0.01 female Mecp2+/- vs. Mecp2+/+ rats”. We also have changed “**p < 0.01 vs. female Mecp2+/+ rats” to “**p < 0.01 female Mecp2+/- vs. Mecp2+/+ rats”.

Point 8: Page 20 

Line 385 Change “demonstrate” to “demonstrated”.

Answer 8: Thank you for the correction. We have accordingly changed “demonstrate” to “demonstrated”.

Point 9: Abbreviations Place the abbreviations in alphabetical order.

Answer 9: Thank you for the correction. In accordance, we have replaced the abbreviations in alphabetical order.

 

Reply to reviewer #2

Comments to authors

Point 1: Page 4, Line 78. Remove “on so on”. 

Answer 1: Thank you for the correction. In accordance, we have removed “on so on”.

Point 2: Page 7, Line 150. This is the Figure legend for figure 2, not sure why it is in the methods section.

Answer 2: Thank you for the comment. Since we had mentioned the figure in the materials and methods section “Morris water maze (MWM) test”, the reviewer#1 had advised the figure to be numbered figure 2 in the previous comments, according to the Plos One’s instructions. However, since the mismatch between the order of figures and results could confuse readers, we have replaced the Fig 2 after the result section “Impaired spatial learning and memory in Mecp2-deficient female rats” and altered the order of the figure number.

Point 3: Page 12, Line 260-261. “Both the frequency (Fig 4A) and duration 261 (Fig. 4B) of self-grooming behavior were significantly or tended to be reduced in female Mecp2+/−…” …… The current phrasing is not clear, authors should clarify that frequency was reduced and duration tended to be reduced but not significant.

Answer 3: Thank you for the correction. In accordance, we have revised the sentence to clarify that frequency was reduced and duration tended to be reduced but not significant.

Point 4: Page 12, Line 283. “Spontaneous locomotor activity was significantly or tended to be lower among female Mecp2+/− rats compared to female Mecp2+/+ rats (Fig 5) during both light and dark phases.” Authors should clarify the specific timepoints of significance.

Answer 4: Thank you for the comments. We have shown the locomotor activities at each 30-min from 16 w or 23 w old rats over 2 consecutive 24-periods as supplemental S1 Fig in the revised manuscript. After statistical analyses (Kruskal–Wallis with Dunn’s post hoc multiple comparisons test), we have found no significant difference between genotypes at any time point. These results indicated that the Mecp2 deficiency is not likely to affect circadian rhythms but locomotor activity itself. So, we have additionally described it in the revised manuscript.

Point 5: Page 12, Line 285. For consistency lower case w “29 w”

Answer 5: Thank you for the correction. We have accordingly changed “29W” to “29 w”.

Point 6: Page 13, Line 301. It is unclear why the result section “Impaired spatial learning and memory in Mecp2-deficient female rats” is Figure 2, following Figure 6.

Answer 6: Thank you for the correction. In accordance, we have revised this matter as described in Point 2. 

Point 7: Page 14

Line 314. “Surprising, however, biogenic amine and…” – Change to “Surprisingly”

Line 323. “These results suggest that impaired cholinergic signaling would severely affect the behavioral abnormalities observed...” The word “would” should be changed to “could”.

Answer 7: Thank you for the correction. In accordance, we have changed “Surprising” to “Surprisingly”, and “would” to “could”. 

Point 8: Page 18, Line 344.” Astrocytes are known to support neuronal function....” A reference is needed for this sentence.

Answer 7: Thank you for the correction. In accordance, we have added references for the sentence. 

Point 9: Page 18, Line 355. “Both the neurotrophic factors play important…” – Remove “the”.

Answer 9: Thank you for the correction. In accordance, we have removed “the”.

---

## [Editor Report · Decision Letter 2]

7 Oct 2021

Substantial acetylcholine reduction in multiple brain regions of Mecp2-deficient female rats and associated behavioral abnormalities

PONE-D-21-05550R2

Dear Dr. Fukumitsu,

We’re pleased to inform you that your manuscript has been judged scientifically suitable for publication and will be formally accepted for publication once it meets all outstanding technical requirements.

Kind regards,

Atsushi Asakura, Ph.D

Academic Editor

PLOS ONE
---

## [Editor Report · Acceptance letter]

12 Oct 2021

PONE-D-21-05550R2 

Substantial acetylcholine reduction in multiple brain regions of *Mecp*2-deficient female rats and associated behavioral abnormalities 

Dear Dr. Fukumitsu:

I'm pleased to inform you that your manuscript has been deemed suitable for publication in PLOS ONE. Congratulations! Your manuscript is now with our production department. 

Kind regards, 

on behalf of

Dr. Atsushi Asakura 

Academic Editor

PLOS ONE